# Predicting Maximum Oxygen Uptake from Non-Exercise and Submaximal Exercise Tests in Paraplegic Men with Spinal Cord Injury

**DOI:** 10.3390/healthcare11050763

**Published:** 2023-03-05

**Authors:** Bum-Suk Lee, Jae-Hyuk Bae, Yu-Jin Choi, Jung-Ah Lee

**Affiliations:** 1Department of Rehabilitation Medicine, Catholic Kwandong University, Incheon 22711, Republic of Korea; 2Department of Clinical Rehabilitation Research, Korea National Rehabilitation Research Institute, Seoul 01022, Republic of Korea

**Keywords:** maximum oxygen uptake, prediction equation, non-exercise test, submaximal test, spinal cord injury

## Abstract

This study aimed to develop prediction equations for maximum oxygen uptake (VO_2_max) based on non-exercise (anthropometric) and submaximal exercise (anthropometric and physiological) variables in paraplegic men with a spinal cord injury. All participants were tested on an arm ergometer using a maximal graded exercise test. Anthropometric variables such as age, height, weight, body fat, body mass index, body fat percentage, and arm muscle mass and physiological variables such as VO_2_, VCO_2_, and heart rate at 3 and 6 min of graded exercise tests were included in the multiple linear regression analysis. The prediction equations revealed the following. Regarding non-exercise variables, VO_2_max was correlated with age and weight (equation R = 0.771, R^2^ = 0.595, SEE= 3.187). Regarding submaximal variables, VO_2_max was correlated with weight and VO_2_ and VCO_2_ at 6 min (equation R = 0.892, R^2^ = 0.796, SEE = 2.309). In conclusion, our prediction equations can be used as a cardiopulmonary function evaluation tool to estimate VO_2_max simply and conveniently using the anthropometric and physiological characteristics of paraplegic men with spinal cord injuries.

## 1. Introduction

Individuals with spinal cord injuries (SCIs) have a higher risk of cardiopulmonary and cardiovascular diseases (CVDs) than the general population. These risk factors include reduced physical activity due to wheelchair dependency in activities of daily living, decreased muscle mass and increased body fat, decreased autonomous nervous system function, and decreased functional body control, which acts as a cause of decreased cardiopulmonary function due to SCI [1]. The prevalence of CVD in SCI is associated with the injury’s level and extent. Tetraplegic level injury was associated with a 16% higher risk of all CVDs (coronary artery disease, hypertension, cerebrovascular disease, valvular disease, and dysrhythmias) and a fivefold increased risk of cerebrovascular disease; however, paraplegic subjects had a 70% greater risk of coronary artery disease [2]. The recognition and treatment of CVD is a developing clinical challenge in this population.

Regular aerobic training improves cardiovascular function, aerobic capacity, and exercise tolerance in individuals with an SCI, often resulting in improved independence in the activities of daily living [3]. In order to evaluate the cardiopulmonary function and maintain health, a safe and effective cardiopulmonary function test with the characteristics of paraplegia or quadriplegia is necessary.

Maximum oxygen uptake (VO_2_max) is an important index for predicting cardiovascular mortality [4]. VO_2_max is closely linked to CVD and is an important index for individuals with an SCI who are generally sedentary [2]. 

However, accurately measuring VO_2_max requires a complex gas analysis, maximal graded exercise tests (GXTs), expensive equipment, a long time, and skilled personnel. In particular, measuring the VO_2_max of patients with SCIs with autonomic nervous system abnormalities and upper arm contracture is difficult.

To address these issues, studies continue to develop various exercise prediction models, such as models using non-exercise and submaximal tests to measure VO_2_max indirectly. A non-exercise prediction equation model is an easy, fast, and convenient method that does not require complicated experimental equipment and maximal effort. 

Non-exercise prediction equations are necessary for the elderly and differently abled individuals who face difficulty wearing the measuring VO_2_max with GXT. However, non-exercise prediction equations have low accuracy [5]. 

A submaximal exercise prediction equation model is used with various populations to assess their fitness levels and predict VO_2_max when a maximal test is not possible. The estimation of VO_2_max during submaximal exercise testing involves physiological responses during GXT in which a steady state can be maintained. Physiological variables during submaximal exercise tests are widely used in estimation due to their high accuracy and validity. Over the past decades, many protocols have been developed for this purpose [6]. 

Several studies have used an arm ergometer to estimate VO_2_max for physiological variables [7,8]. These studies showed that the variables for estimating the VO_2_max for general populations were related to anthropometric, physiological, and work output. However, the variables required for the estimation equation of VO_2_max for the general population were different from those for individuals with SCIs. It is necessary to determine the independent variables that affect the estimation of VO_2_max in individuals with an SCI due to their unique characteristics. Goosey-Tolfrey et al. [9] developed an equation to estimate VO_2_peak using a perceived exertion (RPE) rating for individuals with SCIs. RPE has been widely used to develop equations to estimate VO_2_max and has been a valuable indicator for adjusting exercise intensity [10,11]. However, RPE has a high predictive accuracy only at high values [12,13], and hence, RPE is not recommended for estimating VO_2_max in non-athletes. 

In another study, the researchers developed a prediction equation for estimating VO_2_max in individuals with SCIs based on the submaximal 6 min arm test (6 MAT), with 6 min VO_2_ as the estimation parameter [14]. Hol et al. [15] validated the 6 MAT for assessing cardiopulmonary functioning in individuals with SCIs. 

However, the accuracy of estimated VO_2_max values using the variables in individuals with SCIs through a submaximal test was not high, and the variables that affected the VO_2_max were limited. Predictive estimated variables that reflect the different levels of congenital and acquired cardiopulmonary function should be considered to overcome such low accuracy.

As evidenced by the need for data regarding sex, age, body mass index (BMI), type of disability, and training status to estimate VO_2_max in individuals with an SCI, anthropometric parameters are closely related to VO_2_max [16]. Thus, developing a more accurate and valid VO_2_max estimation equation using anthropometric parameters and metabolic responses is both highly practical and essential. 

Nevertheless, studies attempting to develop equations for VO_2_max estimation using anthropometric and physiological variables in people with SCIs are lacking. Such equations would enable a simpler and more accurate assessment of cardiopulmonary function in the SCI population. This study aimed to develop and validate an equation for VO_2_max estimation using anthropometric parameters and another using both anthropometric and physiological parameters. 

## 2. Methods

### 2.1. Study Population

Thirty-three men were recruited through the Korea National Rehabilitation Center (KNRC). The inclusion criteria were as follows: individuals with SCIs for at least 12 months since onset, an SCI between levels T2 and L5, no history of taking medications that affect blood pressure (BP) or heart rate (HR), no history of orthopedic surgery in the upper arm within 6 months of the study, and no neurological diseases other than the SCI. 

Twenty-six participants were randomly assigned to the estimation group to develop an equation for VO_2_max estimation. Seven participants were randomly assigned to the validation groups for cross-validation. The sample size was selected by referring to the previous literature, where the validation group’s sample size was approximately 20% of the validation group [14]. Furthermore, as a result of the Mann–Whitney U test, there was no significant difference between the anthropometric and physiological parameters between the two groups (Table 1). The American Spinal Injury Association Impairment Scale (AIS) determines neurological levels and SCI severity. Regarding the level of injury, 20 participants with thoracic SCI and 6 participants with lumbar level were assigned to the estimation group, and 6 participants with thoracic SCI and 1 participant with lumbar level were assigned to the validation group. The study participants were divided into the estimation group (*n* = 9) and the validation group (*n* = 2), with all participants having a spinal cord injury (SCI) above the T6 level. Of the total participants, 21 were assigned to the estimation group and had varying degrees of motor completeness (AIS grades A and B) and motor incompleteness (AIS grades C and D), while the remaining 7 were assigned to the validation group and had motor complete SCIs (AIS grades A). 

Table 1 shows the participants’ anthropometric characteristics, which include their age, height, weight, BMI, body fat percentage, and arm muscle mass, as well as their physiological characteristics, such as VO_2_, VCO_2_, HR, and VO_2_max.

This study was approved by the Institutional Review Board of the NRC (IRB no. NRC-2020-02-023), and informed consent was obtained from all participants. 

### 2.2. Experimental Procedure

The participants were asked not to drink alcohol or engage in vigorous physical activity for 48 h before the exercise test and not to smoke tobacco or drink coffee for 3 h before the exercise test. All participants completed the Physical Activity Readiness Questionnaire (PAR-Q+) pertaining to their health status and readiness for physical activity, and then anthropometric variables, including age, height, weight, BMI, body fat percentage, arm muscle mass, resting HR, and resting BP, were measured. Furthermore, we asked the participants to empty their bladders before the assessment to minimize autonomic dysreflexia. A 5 min warmup was performed before the maximal GXT, and participants were allowed to rest until their heart rate returned to normal. Maximal GXT was performed to measure respiratory gases after the resting heart rate was checked. The maximal GXT protocol involved a starting workload of 25 W and 70 RPM with 3 W increments/min. The participants were instructed to maintain a rate of 70 RPM. The exercise was terminated upon meeting three of the following four conditions: (1) VO_2_ plateaued, with no increase in uptake even with an increasing load; (2) a respiratory exchange ratio (RER) of 1.1 or more; (3) an RPE of 17 or more; (4) an HR of 90% or more of the participant’s estimated maximum (220-age). Furthermore, testing was terminated below 60 RPM regardless of the other criteria. The physiological parameters, including VO_2_, VCO_2_, HR, and RER, were measured during GXT. Following the assessment, BP was measured using a cuff (Omron HEM 6232T, OMRON Healthcare Co., Muko, Japan), and the participants rested until their BP returned to baseline. 

### 2.3. Measurements

The participant’s height was measured using a tape ruler from the top of the head to the heels, with the participants lying on a mat with their legs extended. Body weight was measured by weighing the participants with their wheelchairs using a wheelchair scale and then weighing the wheelchair separately. Body fat percentage and arm muscle mass data were obtained using a body composition analyzer (Inbody S 10, Inbody, Seoul, Republic of Korea). Arm muscle mass was used for the analysis by adding the muscle mass of both arms. VO_2_max was measured during an incremental arm crank test to exhaustion. The participants performed the maximal GXT using a wall-mounted arm ergometer (Angio with electrically adjustable wall fixation, Lode, Groningen, The Netherlands). The ergometer was mounted in such a way that the crank axis was at the participant’s shoulder level, so that the participant’s elbow would be slightly bent when the arm was farthest away from the crank handle (Figure 1). Participants with low grip strength wore holding gloves to fix their hands onto the handles. Resting and exercise HRs were continuously measured with Polar H10 (Polar Electro., Oy, Kempele, Finland) attached to the participant’s chest, and respiratory gas data were collected during maximal GXT using K42 b (K42 b, COSMED, Rome, Italy).

### 2.4. Data Analysis

#### 2.4.1. Multiple Regression Analysis

Data were analyzed using SPSS version 21.0 (Chicago, IL, USA) software. For the respiratory gas analysis, the breath-by-breath data obtained during the GXT were averaged every 15 s to obtain a more accurate measure of VO_2_ (3 min mark: 2 min 45 s to 3 min; 6 min mark: 5 min 45 s to 6 min). The highest VO_2_ obtained in a 15 s interval during the maximal GXT was considered the VO_2_max [15]. To develop an equation for VO_2_max estimation, we performed a stepwise multiple linear regression analysis. Two regression models with different independent variables were generated. As the independent variables, the first regression model contained only anthropometric parameters, such as age, height, body weight, BMI, body fat percentage, and arm muscle mass. In contrast, the second regression model contained both anthropometric and physiological parameters, such as VO_2_, carbon dioxide output (VCO_2_), and HR at 3 and 6 min, as the independent variables.

#### 2.4.2. Cross-Validation

Regression models for the 26 participants in the estimation group were analyzed, and the estimation equation was cross-validated with the 7 participants in the validation group. The regression models were evaluated using multiple correlation coefficients (R, R^2^), standard estimated error (SEE), F statistic, Durbin–Watson statistic, tolerance, and variance inflation factor (VIF). For the cross-validation, the two estimation equations were used to calculate the VO_2_max estimate for the validation group, and the average values were compared with the measured VO_2_max using the Wilcoxon signed rank test (α = 0.05) and correlation analysis. We also calculated the percentage error.

## 3. Results

### 3.1. Regression Model with Anthropometric Variables

Model 1, which only contained anthropometric variables from the estimation group, is shown in Table 2. Age, height, weight, BMI, body fat, and arm muscle mass were entered as the independent variables, and finally, age and weight were selected. The regression model had a statistically significant fit (*p* < 0.01). It explained 59.5% of the observed variance, and the Durbin–Watson test result was close to 2 at 2.092, confirming the independence of the residuals. There were no outliers in the distance between standard residuals and Cook, and the SEE was 3.187. VIF, an index for multicollinearity, had a value lower than 10, at 1.038 for both variables. The regression model constant was 40.644; the regression coefficients were –0.190 for weight and –0.156 for age. Thus, the equation for VO_2_max estimation using anthropometric parameters is as follows: VO_2_max = −0.190 (weight, kg) − 0.156 (age, years) + 40.644(1)

### 3.2. Regression Model with Anthropometric and Physiological Variables

Model 2, which contained all variables from the estimation group, is shown in Table 3. Age, height, weight, BMI, body fat, arm muscle mass, resting HR, VO_2_, VCO_2_, and HR at 3 and 6 min were entered as independent variables, and finally, weight, VO_2_, and VCO_2_ at 6 min were selected. The regression model had a statistically significant fit (*p* < 0.01) and explained 79.6% of the observed variance. The Durbin–Watson test result was close to 2 at 2.051, confirming the independence of the residuals. There were no outliers in the distance between standard residuals and Cook, and the SEE was 2.309. VIF was less than 10 for all three variables: 4.232 for VO_2_, 4.146 for VCO_2_, and 1.866 for weight. The regression model constant was 27.495, and the regression coefficient was 1.293 for VO_2_ at 6 min, 1.072 for VCO_2_ at 6 min, and 0.473 for weight. Thus, the equation for VO_2_max estimation using anthropometric and physiological parameters is as follows: VO_2_max = 2.359 (6 min VO_2_) − 1.959 (6 min VCO_2_) − 0.166 (weight, kg) + 27.495(2)

### 3.3. Cross-Validation 

We applied the two regression models for VO_2_max obtained from the estimation group to the validation group for cross-validation. With both models, the measured VO_2_max and predicted VO_2_max were not statistically significantly different. There was no correlation in model 1 (R = 0.607), and there was a significant correlation in model 2 (R = 0.893, *p* < 0.01). The percentage error was 1.61 ± 25.73 for model 1 and 1.36 ± 16.45 for model 2 (Table 4).

## 4. Discussion

This study aimed to develop equations for estimating VO_2_max in paraplegic men with SCIs using anthropometric and physiological parameters. We generated two models using non-exercise and submaximal tests. 

The accuracy of the estimation equations in the multiple linear regression analysis was indicated by a high correlation coefficient R and low SEE. The SEE predicts the value of the dependent variable in a given model, that is, the value of VO_2_max in our case. The correlation coefficient represents a linear relationship between the measured and estimated VO_2_max, and both estimated error and variability must be considered because the coefficient of variation may be similar even if the correlation coefficient differs. 

In estimation models 1 and 2 generated in this study: the R was 0.771 and 0.892, respectively; the explanatory powers (R^2^) were 59.5% and 79.6%, respectively; and the SEEs were 3.18 and 2.30, respectively. In studies on people without disabilities, VO_2_max estimation models using submaximal exercise tests had R-values of 0.570–0.819, R^2^-values of 56–91%, and SEE values of 3.34–14.40 [8,17,18,19]. Vinet et al. [20] showed that the estimation model had an R^2^ of 81.0% and a SEE of 0.01, where the participants were wheelchair-dependent athletes. Furthermore, other studies reported an R-value of 0.871–0.893, R^2^-value of 59.0%, and SEE value of 1.44–1.57 [21,22]. Although our sample comprised paraplegic men with SCIs who were not involved in sports, unlike previous studies conducted on healthy people or those with an SCI with experience in sports, our models still had relatively good R-, high R^2^-, and low SEE values. This shows that our VO_2_max estimation equations had high accuracy, suggesting that the predictive variables used to develop these equations were potentially helpful in estimating VO_2_max in paraplegic men with SCIs.

There were no statistically significant differences in the measured VO_2_max and estimated VO_2_max values using the two equations in the validation group, and the percentage error was low. Although model 1 did not show a significant correlation, model 2 showed a high correlation. Based on the validation results, model 2 would be useful, owing to its accurate prediction of VO_2_max in paraplegic men with SCIs, but it would be difficult to utilize model 1 in general practice due to its poor predictive power.

Age, an important factor in model 1, has been chosen in most estimation equations using physical variables [20,21], so it is a crucial factor for VO_2_max estimation without submaximal exercise tests. In model 1, age was negatively correlated with VO_2_max. In a study of differently abled people, increased body fat and reduced muscle mass with aging led to reduced VO_2_max [23]. This is consistent with the results of our study. In other words, as with healthy people, people with an SCI also have challenges in performing high-intensity exercise and experience marked changes in their body composition as they age. Ultimately, age is an important predictor in estimating VO_2_max. Body weight, an important variable in models 1 and 2, is also frequently utilized to estimate VO_2_max [20,21]. Physical ability is generally determined by unmodifiable factors such as degree of lesions, age, and sex, but modifiable factors such as the amount of physical activity and BMI also play a role [24]. Previous studies reported that BMI, an index reflecting height and weight, is also useful as a predictor of VO_2_max [25]. However, our findings suggest that body weight was more important than BMI as a predictor of VO_2_max. 

In people with paraplegia due to an SCI, the lower extremity’s muscle mass varies depending on injury severity. This seems to explain the poor correlation of VO_2_max with BMI and the stronger correlation with primary variables such as weight and age. In light of this, subsequent studies that attempt to develop more precise equations should choose variables reflecting the different characteristics of people with SCIs, including upper extremity muscle mass and power that exclude lower extremity data. 

In model 2, which used both anthropometric and physiological variables as the independent variables, body weight, VO_2_, and VCO_2_ at 6 min were identified as significant factors. VO_2_ was used during submaximal exercise as an important variable in the study on VOmax in individuals with spinal cord injuries [4,14,15,26]. Among these studies, Totosy de Zepetnek et al. [14] and Hol A.T. et al. [15] reported that VO_2_ in the 6 min arm ergometer test was an independent predictor of VOpeak, and the developed equation was cross-validated to produce an accurate estimation of VOpeak in individuals with tetraplegia and paraplegia. Although this study was conducted on paraplegics, VO at 6 min was found to be a significant variable in predicting VOmax, as in previous studies. 

Model 2 excluded HR values at both 3 and 6 min. HR is an important factor in VOmax estimation for healthy people because VO_2_ is linearly correlated with HR. Thus, the resting or maximum HR values can be used to estimate VO2 in this population [18]. However, it has been reported that HR cannot be used as a predictor of fitness or health in people with SCIs due to their autonomic neuropathy [14]. As we included patients with SCIs between levels T2 and L5, nine people with autonomic neuropathy (SCI affecting T6 or higher) were also included. Moreover, we observed that they had irregular heart rates. Ultimately, HR had a relatively lower R^2^-value compared with other physiological parameters. 

The major cardiovascular concerns associated with SCIs are associated with the following: greater morbidity and mortality from cardiovascular causes; heightened cardiovascular risk factors including low high-density lipoprotein cholesterol, high total cholesterol, and low-density lipoprotein; and higher prevalence of obesity and greater visceral adipose tissue [1].

Age and weight were identified as more significant factors than height, BMI, body fat percentage, and arm muscle mass, and we were able to estimate VO_2_max in people with SCIs solely based on age and weight without other physiological data. In the VO_2_max estimation equation using a submaximal exercise test, we estimated VO_2_max in people with SCIs based solely on weight and VO_2_ and VCO_2_ at 6 min. Weight, a factor used in both models, is closely linked to VO_2_max; weight management is essential to improve fitness in people with SCIs.

Although model 1, which only contained anthropometric parameters, had a lower explanatory power and validity than model 2, which contained anthropometric and physiological parameters, one key advantage was that it could estimate VO_2_max in paraplegic men with SCIs based only on the most basic data such as weight and age. In other words, the equation can be used without any temporal or spatial restrictions for estimating a particular group’s fitness level based only on age and weight without requiring special assessments. Submaximal exercise testing was required to estimate VO_2_max with data for VO_2_ and VCO_2_ at 6 min. The test only takes 6 min to estimate VO_2_max. Thus, the submaximal test using weight and VO_2_ and VCO_2_ values at 6 min developed in this study can be used in various settings, such as rehabilitation centers and fitness centers, for a detailed fitness assessment for exercise prescription with only a simple 6 min assessment using an arm ergometer.

## 5. Limitations

The results of this study are difficult to generalize to the SCI population due to the following limitations: the sample size was small with only 26 subjects. Furthermore, only individuals with SCIs between levels T2 and L5 were included in the study. In addition, individuals with an SCI above T6 with a potential autonomic impact as well as smoking were included. Moreover, this study focused on paraplegic men with SCIs and excluded women. Therefore, further studies are needed to develop prediction equations for VO_2_max in paraplegic women. 

## 6. Conclusions

This study developed an equation for VO_2_max estimation using anthropometric parameters and another equation using anthropometric and physiological parameters to present a practical and accurate equation for VO_2_max estimation. In the cross-validation, no statistically significant differences were observed between the measured VO_2_max and predicted VO_2_max with both models. The developed VO_2_max estimation equation may be helpful in practice for formulating exercise plans in health centers for paraplegic men with an SCI.

## Figures and Tables

**Figure 1 healthcare-11-00763-f001:**
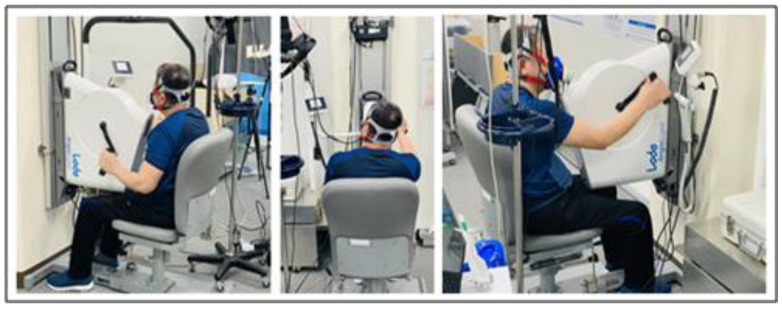
Maximal graded exercise using arm ergometer.

**Table 1 healthcare-11-00763-t001:** Anthropometric and physiological characteristics of participants.

Characteristics	Estimation Group (*n* = 26)	Validation Group (*n* = 7)	*p*-Value
Spinal cord injury level (T/L)	20/6	6/1	
Above T6/below T7	9/17	2/5	
Paraplegia	26	7	
ASIA grade (A/B/C/D)	21/0/2/3	7/0/0/0	
Age (years)	40.77 ± 14.01	47.86 ± 8.05	0.199
Height (cm)	168.48 ± 72.08	169.13 ± 4.51	0.914
Weight (kg)	72.08 ± 13.70	69.27 ± 11.97	0.651
BMI (cm/kg)	25.07 ± 3.89	24.26 ± 4.86	0.531
Body fat (kg)	25.75 ±8.00	23.89 ± 11.69	0.424
Arm muscle mass (kg)	6.52 ± 1.23	7.01 ± 1.09	0.308
Resting HR (bpm)	77.27 ± 10.87	78.14 ± 8.57	0.880
VO_2_ _3 min_ (mL/min/kg)	12.01 ± 2.37	11.28 ± 2.06	0.476
VCO_2_ _3 min_ (mL/min/kg)	11.54 ± 2.27	11.05 ± 2.06	0.651
HR_3 min_ (bpm)	116.52 ± 19.47	115.42 ± 10.57	0.846
VO_2_ _6 min_ (mL/min/kg)	14.06 ± 2.63	12.99 ± 1.34	0.399
VCO_2_ _6 min_ (mL/min/kg)	14.34 ± 2.63	13.42 ± 1.57	0.375
HR_6 min_ (bpm)	128.98 ± 22.70	123.62 ± 13.06	0.424
VO_2_max (mL/min/kg)	20.61 ± 4.80	20.65 ± 4.89	0.747

Data are presented as means ± standard deviations, *p*-value < 0.05. Spinal cord injury (T/L): thoracic/lumbar level, ASIA: American Spinal Cord Injury Association, BMI: body mass index, HR: heart rate, VO_2_
_3 min_: 3 min point oxygen consumption, VCO_2_
_3 min_: 3 min point carbon dioxide production, VO_2_
_6 min_: 6 min point oxygen consumption, VCO_2_
_6 min_: 6 min point carbon dioxide production, HR _3 min_: 3 min point heart rate, HR _6 min_: 6 min point heart rate, arm muscle mass: sum of muscle mass of both arms, VO_2_max: maximal oxygen consumption.

**Table 2 healthcare-11-00763-t002:** Multiple linear regression analysis to estimate arm ergometer VO_2_max based on anthropometric variables.

R	R^2^	SEE	F	*p*	Durbin–Watson	
0.771	0.595	3.187	16.864	0.000	2.092	
	Unstandardized Coefficients		Standardized Coefficients	*p*	Collinearity Statistic	
	B	Std.E			Tolerance	VIF
(constant)	40.644	3.627		0.000		
Weight	−0.190	0.047	−0.542	0.001	0.963	1.038
Age	−0.156	0.046	−0.454	0.003	0.963	1.038

**Table 3 healthcare-11-00763-t003:** Multiple linear regression analysis to estimate arm ergometer VO_2_max based on anthropometric and physiological variables.

R	R^2^	SEE	F	*p*	Durbin–Watson	
0.892	0.796	2.309	28.698	0.000	2.051	
	Unstandardized Coefficients		Standardized Coefficients	*p*	Collinearity Statistic	
	B	Std.E			Tolerance	VIF
(constant)	27.495	6.292		0.000		
VO_2_ (6 min)	2.359	0.361	1.293	0.000	0.236	4.232
VCO_2_ (6 min)	−1.959	0.358	−1.072	0.000	0.241	4.146
Weight	−0.166	0.046	−0.473	0.002	0.536	1.866

**Table 4 healthcare-11-00763-t004:** Measured and predicted VO_2_max (mL/min/kg) from maximal graded exercise test.

	Measured VO_2_max	Predicted VO_2_max	
		Anthropometric Variables	Anthropometric and Physiological Variables
S1	24.64	23.52	26.28
S2	23.05	18.98	20.74
S3	25.88	22.48	20.97
S4	21.38	18.85	20.69
S5	14.66	20.74	16.14
S6	21.83	17.75	20.47
S7	13.13	17.81	17.24
Mean	20.65 ± 4.89	20.02 ± 2.28	20.36 ± 3.25
*p*	-	0.866	0.866
% error		−1.61 ± 25.71	−1.36 ± 16.45
R		0.607	0.893 **

** *p* < 0.01.

## Data Availability

The data presented in this study are available on request from the corresponding author.

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
