# Peer review of "Predicting Maximum Oxygen Uptake from Non-Exercise and Submaximal Exercise Tests in Paraplegic Men with Spinal Cord Injury"

_healthcare, 2023, doi:10.3390/healthcare11050763_

Round 1

Reviewer 1 Report

The authors conclude that their prediction model can be used as a cardiopulmonary function evaluation tool to estimate VO2max using anthropometric and physiological characteristics of paraplegic men with spinal cord injury.

The manuscript is well structured and scientifically sound. The relevant literature is adequately presented and also well discussed. Nevertheless, I have some relevant comments/suggestions:

1.) As personalized medicine becomes increasingly important and also women with SCI might profit from such an investigation, the authors might comment on the fact why only men had been included into this study?  

2.) Was there any follow-up about these patients with regard to cardiovascular outcome? Was the estimated VO2 predictive for adverse cardiovascular events in SCI patients?

Author Response

  1. As personalized medicine becomes increasingly important and also women with SCI might profit from such an investigation, the authors might comment on the fact why only men had been included into this study?  

Authors’ Response: In this study we focused on paraplegic men with spinal cord injury; however, we agree that an investigation for women with SCI is also very important. Therefore, we have added this as a limitation of our study and believe future studies need to be conducted on women as well.

  1. Was there any follow-up about these patients with regard to cardiovascular outcome? Was the estimated VO2 predictive for adverse cardiovascular events in SCI patients?

 Authors’ Response:  Yes, we did follow-up patients with regard to cardiovascular outcomes and also conducted cross-validations. Please refer to Table 4.

Reviewer 2 Report

Thank you for submitting your work to Health Care. Below I have provided a point by point review of each section followed by a general review of each section and/or the overall paper. My goal is to provide feedback that I believe will help improve the paper while maintaining the rigors of science and the Journal. I hope you find the review helpful, and thank you for your hard work.

Abstract:

Introduction:

Line 27-28: This is highly dependent on the injury type and severity. This must be addressed or the introduction remains incredibly ambiguous

Line 31-32: You have to complete a fitness test to be physically active or maintain health?

Line 35: requires citation

Line 38-40: Yes, maximal VO2 testing in individuals living with SCI is difficult, but arm fatigue is a component of maximal testing if using an arm ergometer

Line 65-66: requires citation. RPE does not always correlation with certain exercising variables in SCI

The introduction is a bit convoluted and repeats itself in the second half. There is no need to repeat some information and a significant portion of the introduction can be removed. Accuracy and additional citations are required.

Methods:

 Inclusion criteria drastically limits generalizability in the SCI population considering the number of individuals with higher injuries.

Line 95: I don’t understand your sample size in reference to the cross-over. Please clarify

Line 101: Smoking was not an exclusion criteria? This should be clarified. Were there differences between smokers and non-smokers?

Line 106-108: Due to your injury inclusion criteria, how many individuals are above t6 to where autonomic dysreflexia is a concern?

Line 110-114: There were actually 5 conditions if you include the RPM criteria

Line 146-147: If the W are changing every minute, why are you taking the first 15sec of the minute? Did you average your variables at baseline, during the warm-up and then during exercise? This is unclear. For example, when determining VO2 max why would you not use the final 15s of the completed stage to the final 15s of exercise to determine if a change in VO2 did or did not occur?

The validation group is not explained or justified anywhere in the methods and remains ambiguous. How did you pick 7 individuals for the cross validation? What did they do differently? It appears you simply ran 33 exercise tests and arbitrarily labeled 7 to a cross validation group. Clarity is required.

You have not clearly labelled your dependent variable and at what timepoint this was reached. By the title I assume it is max VO2, but this still needs to be stated.

Results:

For the populations, there is virtually no information presented about the lesion and severity of the lesion which is not appropriate. These data need to be presented in order to determine the relevance of the outcomes. For example, you cite the Hol article, you should have similar information as they providing in their table 2.

Discussion

 The Vinet paper included more than just SCI and within SCI, they parsed out their group based on high and low lesions for their equations which confounds the comparison with your data and this should be mentioned.

Comparison the Hol paper is also confounded by your selection criteria. The Hol paper included primarily AIS A and B participants and 17/13 tetraplegia/paraplegia. This drastically impacts your comparison since none of your participants would have upper extremity impairments.

Your statements about the using HR to predict VO2 max in SCI is also inaccurate. You cite based on their neuropathy, but this is dependent on injury level, which you have potentially excluded from your study population. Based on your reporting you could have 1 individual with an injury at T3 and the rest at T11 which would nullify much of your justification. Moreover, the predominant group in these other studies are tetraplegia which continues to confound your comparisons. Also note, that the citation used for this statement used the data from the Hol study.

Figures/Tables

Table 1: Are you using the suprascript a to demarcate significance or mean +/- SD? If it is mean +/-SD you can just say the table is reported in those units, if not all the text would need a suprascript a.

Also, is the Pvalue presented a comparison of all 3 groups in the table? This is unclear.

Overall Impression:

 The introduction needs significant work on syntax and repetitive information. The introduction could be significantly shorter and more succinct. Regardless, the introduction requires improvements in the citations reported and clarification based on injury levels used in these studies. The cross-validation section was not explained and remains unclear. There is virtually no information reported about the injury level of these participants confounding the comparison to the heavily cited papers. Without clear information about the injury level as well as how this impacts your findings compared to the clearly different study populations of the cited literature, it is impossible to determine the usefulness of these results.

Much of the concerns with the paper could be easily answered with additional information about the participants and how it impacts the data. As written the ambiguity makes it difficult to interpret the importance of the paper.

Author Response

  1. Line 27-28: This is highly dependent on the injury type and severity. This must be addressed or the introduction remains incredibly ambiguous

 Authors’ Response: We have revised the text to address this issue.

 “These risk factors include reduced physical activity due to wheelchair dependency in activities of daily living, decreased muscle mass and increased body fat, decreased autonomous nervous system function, and decreased functional body control, which acts as a cause of decreased cardiopulmonary function due to SCI [1]. Prevalence of CVD in SCI was associated with both the level and extent of injury. Tetraplegic level injury was associated with a 16% higher risk of all CVDs (coronary artery disease, hypertension, cerebrovascular disease, valvular disease, and dysrhythmias) and a fivefold increased risk of cerebrovascular disease, but paraplegic subjects had a 70% greater risk of coronary artery disease [2]. The recognition and treatment of CVD is a developing clinical challenge in this population.”

  1. Line 31-32: You have to complete a fitness test to be physically active or maintain health?

Authors’ Response: We have revised the text to clarify this issue.

 “Regular aerobic training improves cardiovascular function, aerobic capacity, and exercise tolerance in individuals with a SCI, often resulting in improved independence in activities of daily living [3]. In order to evaluate such cardiopulmonary function and maintain health, a safe and effective cardiopulmonary function test with the characteristics of paraplegia or quadriplegia is necessary.”

  1. Line 35: requires citation

Authors’ Response: We have added the reference.

  1. Line 38-40: Yes, maximal VO2 testing in individuals living with SCI is difficult, but arm fatigue is a component of maximal testing if using an arm ergometer

Authors’ Response: We have revised accordingly.

  1. Line 65-66: requires citation. RPE does not always correlation with certain exercising variables in SCI

Authors’ Response: We meant that RPE is a meaningful variable for measuring VO2max. We have revised and added references accordingly.

 “RPE has been widely used to develop equations to estimate VO₂max and has been a valuable indicator for adjusting exercise intensity [10, 11].”

  1. The introduction is a bit convoluted and repeats itself in the second half. There is no need to repeat some information and a significant portion of the introduction can be removed. Accuracy and additional citations are required.

Authors’ Response: We have revised the Introduction section accordingly.

  1. Inclusion criteria drastically limits generalizability in the SCI population considering the number of individuals with higher injuries.

Authors’ Response: We have mentioned this as a limitation of our study in the Limitations Section.

 “The results of this study are difficult to generalize to the SCI population due to the following limitations: the sample size was small with only 26 subjects. Furthermore, only individuals with SCIs between levels T2 to L5 were included in the study.”

  1. Line 95: I don’t understand your sample size in reference to the cross-over. Please clarify

Authors’ Response: In light of your comment we have explained this point in detail for clarification.

 “Twenty‑six and seven participants were randomly assigned to the estimation and cross‑validation groups, respectively. The sample size was selected by referring to previous literature, where the cross-validation group sample size was approximately 20% of the validation group [14].”

  1. Line 101: Smoking was not an exclusion criteria? This should be clarified. Were there differences between smokers and non-smokers?

Authors’ Response: We did not use smoking as an exclusion criterion. Cardiopulmonary function is influenced by many factors such as obesity, alcohol intake, level of smoking, exercise, outdoor activity and daily sleep (1-4). We did not have a large enough sample size to consider these factors; therefore, they were excluded. However, referring to previous studies (5-7), we set the limiting standards for these factors.

(1) Suminski, R.R.; Wier, L.T.; Poston, W.; Arenare, B.; Randles, A.; Jackson, A.S. The effect of habitual smoking on measured and predicted VO2max. J Phys Act Health 2009, 6, 667-673;, doi: https://doi.org/10.1123/jpah.6.5.667

(2) Kobayashi, Y.; Takeuchi, T.; Hosoi, T.; Loeppky, J.A. Effects of habitual smoking on cardiorespiratory responses to sub-maximal exercise. J Physiol Anthropol Appl Human Sci 2004, 23, 163-169, doi: https://doi.org/10.2114/jpa.23.163

(3) Montoye, H.J.; Gayle, R.I.; Higgins, M.I. Smoking habits, alcohol consumption and maximal oxygen uptake. Med Sci Sports Exerc 1980, 12, 316-321.

(4) Fan, L.M.; Collins, A.; Geng, L.; Li, J.M. Impact of unhealthy lifestyle on cardiorespiratory fitness and heart rate recovery of medical science students. BMC public health 2020, 20, 1-8, doi: https://doi.org/10.1186/s12889-020-09154-x

(5) Ekblom‐Bak, E.; Björkman, F.; Hellenius, M.L.; Ekblom, B. A new submaximal cycle ergometer test for prediction of VO2max. Scand J Med Sci Sports 2014, 24, 319-326, doi: https://doi.org/10.1111/sms.12014

(6) Totosy de Zepetnek, J.O.; Au, J.S.; Hol, A.T.; Eng, J.J.; MacDonald, M.J. Predicting peak oxygen uptake from submaximal exercise after spinal cord injury. Appl Physiol Nutr Metab 2016, 41, 775-781, doi: https://doi.org/10.1139/apnm-2015-0670

(7) Arabi, H.; Vandewalle, H.; Pitor, P.; De Lattre, J.; Monod, H. Relationship between maximal oxygen uptake on different ergometers, lean arm volume and strength in paraplegic subjects. Eur J Appl Physiol Occup Physiol 1997, 76, 122-127, doi: https://doi.org/10.1007/s004210050223

  1. Line 106-108: Due to your injury inclusion criteria, how many individuals are above t6 to where autonomic dysreflexia is a concern?

Authors’ Response: We have added details regarding this for clarification in table 1.

  1. Line 110-114: There were actually 5 conditions if you include the RPM criteria

Authors’ Response: We have revised the text in light of your observation.

 “The exercise was terminated upon meeting three of the following four conditions: 1) VO₂ plateaued, with no increase in uptake even with increasing load, 2) a respiratory exchange ratio (RER) of 1.1 or more, 3) an RPE of 17 or more, 4) a HR of 90% or more of the participant’s estimated maximum (220-age). Furthermore, testing was terminated below 60 RPM regardless of the other criteria.”

  1. Line 146-147: If the W are changing every minute, why are you taking the first 15sec of the minute? Did you average your variables at baseline, during the warm-up and then during exercise? This is unclear. For example, when determining VO2 max why would you not use the final 15s of the completed stage to the final 15s of exercise to determine if a change in VO2 did or did not occur?

Authors’ Response: In previous studies, data was analyzed at intervals of 15 seconds [15]. In this study, 15-second intervals were averaged to analyze changes in VO2 in more detail. The highest VO2 among the averaged data at 15-second intervals was considered as the VO2max. We have revised the text to increase clarity in light of your comments.

 “For respiratory gas analysis, the breath-by-breath data obtained during the GXT was aver-aged every 15 seconds to obtain a more accurate measure of VO2(3 min mark: 2 min 45 s to 3 min, 6 min mark: 5 min 45 s to 6 min). The highest value of VO2 obtained in a 15-second interval during the maximal GXT was considered to be the VO2max.”

Data was not acquired during the warm up exercise. A 5 min warm up for adaption was performed prior to maximal GXT, followed by sufficient rest until the heart rate returned to normal. Then, maximal GXT was performed. We have revised the text to increase clarity in light of your comments.

 “A 5 min of warmup was performed before the maximal GXT, and participants were allowed to rest until their heart rate returned to normal. Maximal GXT was performed to measure respiratory gases after the resting heart rate was checked.”

  1. The validation group is not explained or justified anywhere in the methods and remains ambiguous. How did you pick 7 individuals for the cross validation? What did they do differently? It appears you simply ran 33 exercise tests and arbitrarily labeled 7 to a cross validation group. Clarity is required.

Authors’ Response:  In previous studies related to cross-validation, 10 out of 52 (approximately 20%) and about 25% were picked as cross-validation groups. Hence, we also randomly selected 7 out of 33 (approximately 20%) as a cross-validation group according to previous study. The two groups had no significant differences between them regarding age, height, weight, BMI, arm muscle mass, body fat, and respiratory variables at 3 and 6 min. We have revised the text in the manuscript for reader clarification.

“Twenty‑six and seven participants were randomly assigned to the estimation and cross‑validation groups, respectively. The sample size was selected by referring to previous literature, where the cross-validation group sample size was approximately 20% of the validation group [14]. Furthermore, as a result of the Mann-Whitney U test, there was no significant difference between anthropometric and physiological parameters between the two groups (Table 1).”

  1. You have not clearly labelled your dependent variable and at what timepoint this was reached. By the title I assume it is max VO2, but this still needs to be stated.

Authors’ Response: We revised the text accordingly.

  1. For the populations, there is virtually no information presented about the lesion and severity of the lesion which is not appropriate. These data need to be presented in order to determine the relevance of the outcomes. For example, you cite the Hol article, you should have similar information as they providing in their table 2.

Authors’ Response: We have revised the text in light of your comments.

  1. The Vinet paper included more than just SCI and within SCI, they parsed out their group based on high and low lesions for their equations which confounds the comparison with your data and this should be mentioned.

Authors’ Response: We have corrected the sentence to clarify any confusion.

Vinet et al. [20] showed that the estimation model had an R2 of 81.0% and an SEE of 0.01, where the participants were wheelchair-dependent athletes.

  1. Comparison the Hol paper is also confounded by your selection criteria. The Hol paper included primarily AIS A and B participants and 17/13 tetraplegia/paraplegia. This drastically impacts your comparison since none of your participants would have upper extremity impairments.

Authors’ Response: We referred other references and revised the text for better clarification.

In model 2, which used both anthropometric and physiological variables as the in-dependent variables, body weight, VO₂, and VCO₂ at 6 min were identified as significant factors. VO2 during submaximal exercise is used as an important variable in study on VO₂max in individuals with spinal cord injuries [4, 14, 15, 26]. Among these studies, Totosy de Zepetnek et al [14] and Hol A, T, et al., [15] reported that VO₂ in the 6 min arm ergometer test was an independent predictor of VO₂peak and the developed equation was cross validated to produce an accurate estimation of VO₂peak in individuals with tetraplegia and paraplegia. Although this study was conducted on paraplegics, VO₂ at 6 min was found to be a significant variable in predicting VO₂max, as in previous studies. Furthermore, through the results of this study, it was proved that VO2 at 6 min is more meaningful than VO2 at 3 min during submaximal exercise.

  1. Your statements about the using HR to predict VO2 max in SCI is also inaccurate. You cite based on their neuropathy, but this is dependent on injury level, which you have potentially excluded from your study population. Based on your reporting you could have 1 individual with an injury at T3 and the rest at T11 which would nullify much of your justification. Moreover, the predominant group in these other studies are tetraplegia which continues to confound your comparisons. Also note, that the citation used for this statement used the data from the Hol study.

Authors’ Response: We have conceded that the other studies were conducted in individuals with tetraplegia and paraplegia; however, although this study was conducted on paraplegics, VO₂ at 6 min was found to be a significant variable in predicting VO₂max, as in previous studies.  We have revised the text to clarify this point.

 “Among these studies, Totosy de Zepetnek et al [14] and Hol A, T, et al., [15] reported that VO₂ in the 6 min arm ergometer test was an independent predictor of VO₂peak and the developed equation was cross validated to produce an accurate estimation of VO₂peak in individuals with tetraplegia and paraplegia. Although this study was conducted on paraplegics, VO₂ at 6 min was found to be a significant variable in predicting VO₂max, as in previous studies. Furthermore, through the results of this study, it was proved that VO2 at 6 min is more meaningful than VO2 at 3 min during submaximal exercise.”

  1. Table 1: Are you using the suprascript a to demarcate significance or mean +/- SD? If it is mean +/-SD you can just say the table is reported in those units, if not all the text would need a suprascript a. Also, is the P value presented a comparison of all 3 groups in the table? This is unclear.

Authors’ Response: We have revised the table in accordance with your comment.

Round 2

Reviewer 2 Report

Thank you for submitting your work to Health Care. Below I have provided a point by point review of each section followed by a general review of each section and/or the overall paper. My goal is to provide feedback that I believe will help improve the paper while maintaining the rigors of science and the Journal. I hope you find the review helpful, and thank you for your hard work.

Abstract:

Introduction:

 Introduction is still repetitive, but the opening is more cohesive for the totality of the paper.

Methods:

As written, I don’t think I could replicate the cross-validation group, so this remains unclarified in this version. This would likely clarify some of the following questions.

Is the cross-validation estimating the data obtained, or the estimation group?

Due to the number of individuals above T6, analysis of these individuals should be included, at least in the text. This could be added to table 1 if the format presented below is adopted.

Results:

 If the data in table 1 is the estimated VO2, but you conducted an actual max test, why are the max numbers not presented? I would imagine that the table would present the total group data at VO2 max, then what is estimated and then what is cross-validated. The table provided is rather confusing, I think this is what you’re showing, but it is not clearly labelled.  If this is the case, I would suggest reporting all of your obtained data in one table, and then your sub-samples based on the estimations used in the current table. You would then be able to show that the estimation and cross-validation regressions are not different from each other, and not different that the actually obtained data. This also removes any ambiguity on the measurements.

Also, for the cross-validation, shouldn’t you be showing that your estimation data is “valid” as well as the cross-section estimation data valid to both the estimation and the full group? Without showing no differences compared to the “estimation” data (n = 26) and the total data, I am unsure of the “cross-validation” data is actually valid.

Discussion

Limitations should mention the potential autonomic impact directly as well as smoking.

Line 280 (PDF): nothing was proved in this study.

Figures/Tables

 More on table 1, you have changed to column heading to “estimation”, so are you stating you estimated their age, for example? I’m assuming the estimated variables are the physiological variables at the end of the table. This requires clarification.

Overall Impression:

I understand there is a limited ability to make any claims, but I do think it’s worth it to make note of any differences in responses for those with injuries above T6. Exercising autonomic control of cardiovascular function is still incredibly vague, and any information that can contribute to this area, I believe, is worth mentioning. Based on the AIS scores and samples of individuals above T6, it would seem you have some data that you could mention.  This could be expanded, briefly, following lines 286-289 (in PDF).

Overall, I believe the data required is included, but remains ambiguous. I do not think I could replicate the study as presented causing reservation on my interpretation of these data. However, if this could be clarified, the importance of predicted equations for this group, and research purposes is valuable. Lastly, maintaining a clear distinction on the population of individuals with SCI that this is useful for remains a priority and can be completed with slight improvements to the limitations section.

Author Response

  1. Introduction is still repetitive, but the opening is more cohesive for the totality of the paper.

Authors’ Response : Thank you for your valuable comment. We have revised the Introduction accordingly.

  1. Is the cross-validation estimating the data obtained, or the estimation group?

Authors’ Response : The cross-validation is estimating the data obtained. We have changed the naming of the cross-validation group to the validation group in order to avoid any confusion (Table 1). 

  1. Due to the number of individuals above T6, analysis of these individuals should be included, at least in the text. This could be added to table 1 if the format presented below is adopted.

Authors’ Response : Thank you for your valuable suggestion. We have revised it accordingly.

  1. If the data in table 1 is the estimated VO2, but you conducted an actual max test, why are the max numbers not presented? I would imagine that the table would present the total group data at VO2 max, then what is estimated and then what is cross-validated. The table provided is rather confusing, I think this is what you’re showing, but it is not clearly labelled. If this is the case, I would suggest reporting all of your obtained data in one table, and then your sub-samples based on the estimations used in the current table. You would then be able to show that the estimation and cross-validation regressions are not different from each other, and not different that the actually obtained data. This also removes any ambiguity on the measurements.

Also, for the cross-validation, shouldn’t you be showing that your estimation data is “valid” as well as the cross-section estimation data valid to both the estimation and the full group? Without showing no differences compared to the “estimation” data (n = 26) and the total data, I am unsure of the “cross-validation” data is actually valid.

Authors’ Response : Thank you for your detailed and valuable comment. We have revised in the Methods section accordingly. Moreover, we have revised Table 1 to avoid any confusion.

  1. Limitations should mention the potential autonomic impact directly as well as smoking.

Authors’ Response : Thank you for your valuable comment. We have revised it accordingly.

  1. Line 280 (PDF): nothing was proved in this study.

Authors’ Response : Thank you for your valuable comment. We have deleted the sentence to avoid any confusion.

  1. More on table 1, you have changed to column heading to “estimation”, so are you stating you estimated their age, for example? I’m assuming the estimated variables are the physiological variables at the end of the table. This requires clarification.

Authors’ Response : Thank you for your valuable comment. We have revised it for clarification.

  1. I understand there is a limited ability to make any claims, but I do think it’s worth it to make note of any differences in responses for those with injuries above T6. Exercising autonomic control of cardiovascular function is still incredibly vague, and any information that can contribute to this area, I believe, is worth mentioning. Based on the AIS scores and samples of individuals above T6, it would seem you have some data that you could mention. This could be expanded, briefly, following lines 286-289 (in PDF).

Authors’ Response :  Thank you for your comment. We have mentioned that in the Discussion section.